# Experimental Evolution of West Nile Virus at Higher Temperatures Facilitates Broad Adaptation and Increased Genetic Diversity

**DOI:** 10.3390/v13101889

**Published:** 2021-09-22

**Authors:** Rachel L. Fay, Kiet A. Ngo, Lili Kuo, Graham G. Willsey, Laura D. Kramer, Alexander T. Ciota

**Affiliations:** 1Department of Biomedical Sciences, State University of New York at Albany School of Public Health, Rensselaer, NY 12144, USA; rachellfay@gmail.com (R.L.F.); laura.kramer@health.ny.gov (L.D.K.); 2The Arbovirus Laboratory, Wadsworth Center, New York State Department of Health, Slingerlands, NY 12159, USA; kiet.ngo@health.ny.gov (K.A.N.); lili.kuo@health.ny.gov (L.K.); 3Division of Infectious Diseases, Wadsworth Center, New York State Department of Health, Albany, NY 12208, USA; graham.willsey@health.ny.gov

**Keywords:** West Nile virus, viral evolution, climate change

## Abstract

West Nile virus (WNV, *Flaviviridae*, Flavivirus) is a mosquito-borne flavivirus introduced to North America in 1999. Since 1999, the Earth’s average temperature has increased by 0.6 °C. Mosquitoes are ectothermic organisms, reliant on environmental heat sources. Temperature impacts vector–virus interactions which directly influence arbovirus transmission. RNA viral replication is highly error-prone and increasing temperature could further increase replication rates, mutation frequencies, and evolutionary rates. The impact of temperature on arbovirus evolutionary trajectories and fitness landscapes has yet to be sufficiently studied. To investigate how temperature impacts the rate and extent of WNV evolution in mosquito cells, WNV was experimentally passaged 12 times in *Culex tarsalis* cells, at 25 °C and 30 °C. Full-genome deep sequencing was used to compare genetic signatures during passage, and replicative fitness was evaluated before and after passage at each temperature. Our results suggest adaptive potential at both temperatures, with unique temperature-dependent and lineage-specific genetic signatures. Further, higher temperature passage was associated with significantly increased replicative fitness at both temperatures and increases in nonsynonymous mutations. Together, these data indicate that if similar selective pressures exist in natural systems, increases in temperature could accelerate emergence of high-fitness strains with greater phenotypic plasticity.

## 1. Introduction

Flavivirus is a genus of single stranded, positive sense, enveloped RNA viruses which result in at least 50–100 million infections annually [1]. Several mosquito-borne flaviviruses, including West Nile virus (WNV), Zika virus (ZIKV), and dengue virus (DENV), cause emerging infectious diseases. WNV emerged in the western hemisphere in 1999 and was responsible for an outbreak of neuroinvasive disease. In the following five years, WNV became endemic throughout the U.S. WNV is the most geographically widespread flavivirus and has caused the largest outbreak of neuroinvasive illness in history [2,3]. Since its emergence in 1999, there have been approximately 52,000 WN human cases diagnosed in the U.S., resulting in over 2300 fatalities [4]. WNV lineage 2 has also risen in prevalence in Europe, with an increasing frequency of outbreaks in recent years [5]. WNV is predominately vectored by *Culex* mosquitoes, which are widely distributed, found in temperate regions of Europe, Asia, Africa, Australia, and North, Central, and South America [6,7,8,9]. Several factors have played a role in the emergence of WNV, including increasing globalization and climate change [10]. These factors can alter vector range and facilitate geographic expansion of arboviruses, increasing the public health threat.

The WNV genome is approximately 11 kb in length, containing a single open reading frame composed of three structural and seven nonstructural genes [11,12]. WNV exists within hosts and vectors as a diverse mutant swarm [7,13,14,15,16]. Several studies have demonstrated the importance of WNV minority variants in viral fitness, virulence, and adaptability [7,17,18,19,20]. While the evolutionary trajectory of WNV is known to be influenced by several virus- and host-dependent factors, the role of the environment is not well defined.

The average land and ocean temperatures have increased at a rate of 0.08 °C per decade since 1880. In the last 40 years, the rate has doubled to 0.18 °C, with 2016 and 2020 being the warmest years on record [21]. Temperature has a significant impact on host, vector, and virus interactions which can result in altered transmission of vector-borne pathogens [22]. The majority of the arbovirus life cycle is spent in ectothermic invertebrates which are dependent on external sources of heat for temperature regulation and directly influenced by climate variability. Thermal conditions have been shown to influence vector competence, life history traits, blood feeding behavior, as well as vector, host, and viral distribution [23,24,25,26,27,28]. Documented increases in viral replication resulting from increases in temperature are likely to be coupled to higher mutation frequencies, potentially resulting in accelerated evolutionary rates and altered fitness landscapes [29]. We sought to examine the impacts of temperature on the rate and extent of WNV evolution. Sequential passage was performed in *Culex tarsalis* cells at 25 °C and 30 °C. Full-genome deep sequencing was used to assess genetic signatures during passage, and temperature-dependent alterations to viral fitness were assessed. Together, these data indicate that increases in temperature could accelerate emergence of high-fitness strains.

## 2. Materials and Methods

### 2.1. Viral Strain

The West Nile virus infectious clone (WNV02 IC) was designed from WNV 1986, isolated in 2003 from an American crow in Albany, NY (GenBank: DQ164189.1) [29]. WNV02 IC was developed using modifications of previously described methodology [30]. Low-copy number plasmid vector pACYC177 was modified to include a poly-linker region containing restriction sites BamHI, BsiWI, and XbaI. Four overlapping cDNA fragments encompassing the entire viral RNA genome were generated by reverse transcriptase polymerase chain reaction (RT-PCR) using SuperScript™ III One-Step RT-PCR System with Platinum Taq DNA Polymerase (Thermo Fisher, Waltham, MA, USA), following the manufacturer’s recommended protocol. The final viral cDNA genome was cloned into two separate plasmids designated as pWNV02-5′ half and pWNV02-3′ half (Figure 1). Plasmid pWNV02-5′ half contains the T7 promoter and the 5′-5785 nucleotides of the viral genome that were cloned between BamHI and BsiWI sites, and pWNV02-3′ half contains the 3′ distal 5244 nucleotides of the viral genome between BsiWI and XbaI [31]. Plasmid WNV02 5′ half was digested with BsiWI and plasmid WNV02-3′ half with BsiWI and XbaI (New England Biolabs, Ipswich, MA, USA). The corresponding fragments were purified and in vitro ligated with T4 DNA ligase (New England Biolabs, Ipswich, MA, USA) to generate linearized full-length WNV02 IC. After purification, capped run-off RNA transcripts were synthesized in vitro using HiScrib T7 ARCA mRNA Kit (New England Biolabs, Ipswich, MA, USA) as specified in the manufacturer’s protocol. The in vitro synthesized genomic viral RNA was then transfected into 10^7^ Vero cells (ATCC CCL-81) via electroporation by three pulses at 0.85 kV and 25 μF using a GenePulser apparatus (Bio-Rad Laboratories, Hercules, CA, USA) [31]; transfected cells were resuspended in growth medium containing 10% fetal bovine serum (FBS), plated in a T-25 flasks, and incubated at 37 °C with 5% CO_2_. The flasks were observed daily for signs of infection, and the supernatants (p0) were harvested when more than 70% of cells exhibited cytopathic effect (CPE). The rescued p0 was then amplified once in C6/36 cells (ATCC CRL-1660) to produce p1 virus stock which was used to create viral stocks for experimentation.

### 2.2. Cells and Media

*Culex tarsalis* cells ((C × T) kindly provided by A. Brault, CDC Ft Collins), Vero cells (African green monkey kidney cells, ATCC CCL-81), and Peking duck embryo (ATCC CCL-141) cells were grown at optimum temperature for each cell line, 28 °C and 37 °C, respectively, for insect and vertebrate lines. Confluent monolayers of C × T and PDE cells were prepared for infection by seeding six-well plates (Corning Costar, Cambridge, MA, USA) with 5 × 10^6^ cells/well for C × T cells, 4 × 10^5^ cells/well for PDE cells, or 1 × 10^6^ for Vero cells in 3 mL of appropriate media. Cells were incubated for 4 days for C × T cells and 3 days for PDE and Vero cells. Confluent cell monolayers and virus-infected cells were maintained in the appropriate media for the cell line, Schneider’s 10% FBS for C × T cells, minimal essential medium with 10% FBS for Vero cells, and Eagle’s minimum essential media with 10% FBS for PDE cells.

### 2.3. Serial Passage of Virus and Quantification

Viral serial passage was completed using confluent C × T cell monolayers in 6-well plates infected with two biological replicates at a multiplicity of infection (MOI) of 0.01 with 100 µL of virus. After one hour of absorption at 25 °C or 30 °C, inoculum was removed, washed three times, and overlaid with 3 mL of Schneider’s media. Cell culture supernatants were collected at peak titer, which was 96 h post infection (hpi) based on preliminary studies. WNV02 IC was passaged 12 times at 25 °C and 30 °C. In instances where the desired MOI was not achieved, the maximum viral input was utilized. This occurred only once during passage, with 25 °C lineage A pass 3, where an MOI of 0.008 was used to initiate passage. Samples were stored in 900 µL BA-1 with 20% FBS. RNA was extracted using a MagMAX viral isolation kit (Applied Biosystems, Waltham, MA, USA) on a Tecan Evo 150 liquid handler (Tecan, Mannedorf, Switzerland). Real-time quantitation RT-PCR was completed using TaqMan One-Step RT-PCR master mix (Applied Biosystems, Waltham, MA, USA) and analyzed on Quant Studio 5 (Thermo Fisher, Waltham, MA, USA). WNV primers and probes were designed as previously described and plaque-forming unit (PFU) standards were utilized for quantification [32]. In addition, plaque titration was done using Vero cells to determine viral titer according to the standard protocol [33]. Passage titers at 96 hpi were analyzed using one-way analysis of variance (ANOVA) with multiple comparisons and Tukey’s post-correction. All statistics were performed using GraphPad Prism Version 9.

### 2.4. Growth Kinetics

Viral growth kinetics of both lineages (A/B) of WNV pass 1, 3, 6, and 12 were assessed as described above, with samples from each passage assessed at both 25 °C and 30 °C in C × T cells. In addition, growth kinetics of pass 1 and 12 WNV were assessed in PDE cells at 37 °C. Infections were performed at an MOI of 0.001 in C × T cells and an MOI of 0.01 in PDE cells with 100 µL of virus. Supernatant was sampled every 24 h up to 168 h. Replicative fitness was determined via linear regression analysis of the log linear portion of the growth curve. The slopes of the resultant lines (defined as replicative fitness (log_10_ growth per hour)) were then compared using three-way ANOVA with multiple comparisons and Tukey’s post-correction to indicate statistical significance. All statistics were performed using GraphPad Prism 9.

### 2.5. Sequencing

RNA was extracted using RNeasy purification kit (Qaigen, Hilden, Germany). Complete genomes of samples were amplified into 2 kb fragments with six overlapping WNV primer sets (sequences available upon request) using a SuperScript III One-Step RT-PCR kit (Life Technologies, Carlsbad, CA, USA) and One-Step Ahead RT-PCR kit (Qiagen, Hilden, Germany) [34]. Products were visualized using gel electrophoresis. Samples were prepared for deep sequencing as described previously [7]. Deep sequencing was performed using an Illumia MiSeq platform at the Wadsworth Center Applied Genomics Core. Sequence data were analyzed using Geneious Prime 2020. Reads were paired, merged, and mapped to the reference genome of the WNV02 IC, followed by variant calling of single nucleotide polymorphisms (SNPs) above 2% with minimum depth of 100×. The proportion of nonconsensus nucleotide and amino acid substitutions >2% of sequences was calculated for each passage sequenced and averaged among 12 passages for each temperature. Proportions of nonconsensus reads were compared among temperatures using chi-squared tests and mean mutations/passage were compared with *t*-tests in GraphPad Prism 9. The proportions of nonsynonymous mutations per nonsynonymous site (dN) and synonymous mutations per synonymous site (dS) were calculated in MEGA X [35] and dN/dS ratios were used to evaluate intrahost selective pressures.

## 3. Results

### 3.1. In Vitro Passage of WNV02 at Increased Temperature in Mosquito Cells

Viral loads were quantified at both 25 °C and 30 °C at 24–120 hpi to establish baseline temperature sensitivity and growth kinetics of WNV02 IC in C × T. These temperatures were utilized to assess the influence of temperature within a realistic range. Results demonstrated significantly lower viral titers at 25 °C at 24–96 hpi (* *p* ≤ 0.05 one-way ANOVA w/multiple comparisons, Tukey’s post-test) with similar peak titers by 120 hpi (Figure 2A). To determine the influence of temperature on WNV evolution and adaptation, serial passage of WNV02 IC was performed in C × T mosquito cells at 25 °C or 30 °C using two distinct lineages (A and B) maintained at each temperature. Viral loads, as measured by WNV genomes, varied by passage and temperature (Figure 2B). Specifically, decreased titer was measured for pass 1 and 2 at 25 °C, with substantial increases at pass 5. Output titer from passes 5–12 fluctuated between 9.0 and 11.0 log_10_ genomes/mL. At 30 °C, increased output titers were measured for each passage up to pass 5 and a gradual decrease was measured from pass 7–12. Similar output titers were measured for lineage A and B at each temperature. Overall, a significantly higher viral load was measured for WNV passaged at 30 °C compared to 25 °C (* *p* ≤ 0.05 one-way ANOVA w/multiple comparisons, Tukey’s post-test). Infectious particles were also quantified via plaque assay (pfu). Genome to pfu ratios fluctuated during passage, with values ranging from 6.0 × 10^2^–6.0 × 10^3^ copies/pfu, yet overall trends for infectious particles were similar to those measured for WNV genomes (Figure 2B).

### 3.2. Viral Fitness following Passage at Increased Temperature

Growth kinetics of WNV before (pass 1) and after (pass 12) were quantified in C × T cells at both 25 °C and 30 °C to evaluate the temperature-specific effects of experimental passage on viral replication. To quantify replicative fitness, log linear growth rates were calculated and compared [34]. Increased replicative fitness at both 25 °C and 30 °C was measured for both lineages following passage at 30 °C but only increases at 25 °C were measured in one lineage (lineage B) following passage at 25 °C. Specifically, replicative fitness at matched growth temperature following passage was unchanged in lineage A 25 °C, but increased by 0.034 log_10_ pfu/h in lineage B 25 °C, 0.041 log_10_ pfu/h in lineage A 30 °C, and 0.011 log_10_ pfu/h in lineage B 30 °C. Replicative fitness at alternate growth temperature decreased by −0.002 log_10_ pfu/h in lineage A 25 °C, yet increased by 0.013 log_10_ pfu/h in lineage B 25 °C, 0.012 log_10_ pfu/h in lineage A 30 °C, and 0.027 log_10_ pfu/h in lineage B 30 °C. Combined, replicative fitness at 25 °C and 30 °C was statistically equivalent following a single amplification (pass 1) on C × T at either temperature, yet passage at both 25 °C and 30 °C (pass 12) resulted in increased replicative fitness when growth kinetics were assessed at matched temperatures (Figure 3; *p* ≤ 0.05, three-way ANOVA w/multiple comparisons, Tukey’s post-test). Interestingly, WNV passaged at 25 °C did not demonstrate increased replicative fitness at 30 °C, yet WNV passaged at 30 °C demonstrated significant increases at both temperatures (Figure 3; *p* ≤ 0.05, three-way ANOVA w/multiple comparisons, Tukey’s post-test stat), indicating broad adaptation.

Replicative fitness of C × T passaged WNV was additionally quantified in PDE cells to assess if passage and adaptation to mosquito cells resulted in altered growth kinetics in avian cells. Although differences were not significant (*p* = 0.24), replicative fitness was lower in avian cells following mosquito cell passage at both 25 °C and 30 °C (Figure 4).

### 3.3. Temperature and Lineage Specific Genetic Signatures

Whole genome sequencing of pass 1, 3, 6, and 12 of both lineages and temperature treatments was completed to identify consensus and minority genetic variation resulting from WNV passage. A total of nine nucleotide mutations resulting in six amino acid substitutions were identified in consensus sequences, none of which were shared among lineages or temperatures. Two nonsynonymous mutations in the envelope protein, A1427G (N444S) and A1450G (T452A), in addition to C3815T (A1240V) in the nonstructural protein 2A (NS2A), were identified in lineage B following passage at 25 °C, yet no consensus change was identified in lineage A (Figure 5). Two unique nonsynonymous mutations identified in the envelope protein, T1034C (L313S) and T1682C (L529S), were identified in lineage B following passage at 30 °C, while the nonsynonymous mutation C815T (A240V) in the pre-membrane (prM) protein was identified in lineage A following passage at 30 °C. Three of these mutations, T1682C (L529S), T1034C (L313S), and C815T (A240V) were identified in minority populations of intermediate passages prior to their fixation in consensus sequences (Figure 5A). Additionally, all lineages and temperatures shared one minority mutation, A3352C (T1077P), in the NS1 (Table 1). T3287G (V1064G), T3333G (S1079R), and A3337G (S1081G), in the NS1, were also identified in one or both lineages following passage at both temperatures. C151A in the 5′ UTR and A6347G (E2084G) in the NS3 were also identified in one or both lineages following passage at both temperatures. Two minority changes were shared among lineages but found to be temperature specific. A1142G (Y3049C) in the prM was identified in lineage A and B following passage at 30 °C, and T3234A (N1046K) in the NS1 was identified in lineage A and B following passage at 25 °C (Table 1).

The proportions of nonconsensus nucleotide and amino acid substitutions were calculated for each lineage at pass 1, 3, 6, and 12 to assess genetic diversity and rate of evolution over passage (Figure 6A). Intrahost diversity throughout passage was highly variable, with higher proportions of nonsynonymous substitutions. Passage at 25 °C and 30 °C resulted in similar trends, with a decrease in intrahost diversity from pass 1–3, followed by modest increases at pass 6 and pass 12. Despite fluctuation, mean intrahost diversity was higher in both nucleotide and amino acid levels during passage at 30 °C compared to passage at 25 °C, and these differences were significant for nonsynonymous mutations (Figure 6B; * *p* ≤ 0.05, chi-square test). In order to assess the frequency of unique mutations with potentially increased phenotypic relevance, we quantified new mutations in >5% of sequences and compared among passage temperatures. Overall, the mean number of unique amino acid substitutions identified in viruses passaged at 30 °C was significantly higher than those passaged at 25 °C, 8.0 and 5.25 per passage, respectively (Figure 6C; *p* ≤ 0.05, *t*-test). Calculated dN/dS values were similar among passage temperatures for mutations at frequencies >2% (0.19) but modestly higher for 30 °C populations when considering mutations >5% (dN/dS = 0.96 and 1.15 for 25 °C and 30 °C passaged populations, respectively). 

## 4. Discussion

Temperature has been shown to have a significant influence on arbovirus transmission, which is generally attributed to increased replication rates accelerating dissemination and decreasing extrinsic incubation periods [23,25,26,27,36]. While increased replication should also be coupled with increased mutation frequencies, the influence of temperature on RNA virus evolution and adaption at higher temperatures in mosquitoes has not been adequately investigated. We utilized in vitro passage to test the impact of temperature on West Nile virus evolution in mosquito cells. We found that passage at higher temperatures resulted in a more rapid accumulation of nonsynonymous mutations that facilitated cell-specific adaptation at both 25° and 30 °C, suggesting that, if similar selective pressures exist in mosquitoes, increasing temperatures in nature could facilitate the emergence of broadly adaptive, high fitness WNV strains.

RNA viruses are thought to exist at an error threshold, where increases in mutational load could be detrimental to viral fitness if accumulation of deleterious mutations outpaces purifying selection [37]. Theoretically, increases in temperature could therefore decrease viral fitness, yet since WNV and other arboviruses are additionally required to efficiently replicate at internal temperatures of vertebrate hosts (37–39 °C), our study is well within the thermal limits of replication. Exceeding thermal limits during replication in mosquitoes would likely be quite rare, and detrimental effects on mosquito fitness are much more likely to occur with rising ambient temperatures. Indeed, unimodal relationships based on experimental data for WNV suggest that although transmission efficiency peaks between 23 °C and 26 °C, replication rates in mosquitoes likely peak at temperatures between 35 °C and 40 °C [38]. It is also important to distinguish between mutation frequency (the accumulation of mutations resulting from replication and evolutionary pressures) and mutation rate (the intrinsic error rate of the RNA-dependent RNA polymerase). The influence of temperature on the latter has not been investigated. While different levels of intrahost diversity of WNV have been identified in mosquito and vertebrate hosts, these differences have historically been attributed to host-specific selective pressures [34,39,40].

We identified relatively few consensus changes resulting from passage, which is consistent with past studies utilizing in vitro experimental evolution of WNV to evaluate selective pressures [17,34,41,42]. These studies generally demonstrate that modest consensus changes can often drive highly divergent phenotypic changes. There are also highly relevant examples of this in naturally circulating strains, specifically during the early spread of WNV across the Americas. In the early 2000s, the WNV02 genotype emerged, which is defined by a single shared amino acid change in the envelope protein, V159A [43]. This newly emergent genotype was found to be transmitted earlier and more efficiently by *Culex* spp. Additionally, a substitution in the NS3 helicase, T249P, was found to increase virulence in avian hosts [44]. Similar instances of single substitutions driving epidemiologically relevant phenotypic change have been observed with other arboviruses. A 2005 epidemic of chikungunya virus (CHIKV) was associated with a mutation in the envelope protein which facilitated adaptation to *Aedes albopictus [45]*. Similarly, a mutation in the NS1 protein of ZIKV strains that successfully invaded the Americas was found to promote infectivity and increase prevalence in *Aedes* mosquitoes [46].

Consensus mutations identified in our study were found throughout the WNV genome, with amino acid substitutions identified in the envelope, pre-membrane, and NS2A proteins. NS2A plays a role in viral replication, assembly, and modulation of host antiviral response [47]. Arboviruses elicit an RNA interference (RNAi) response in mosquitoes during infection and recent data suggest NS2A acts as a viral suppressor of RNAi by direct binding and sequestering of RNAi complexes [48,49,50]. RNAi has been shown to be influenced by temperature, with reduced activity at cooler temperatures [51]. Given that A1240V in NS2A was associated with temperature-specific adaptation at 25 °C, it is feasible that this residue could have a role in RNAi suppression. The envelope protein aids in virion assembly, receptor binding, and pre-membrane fusion [48]. The pre-membrane protein functions as a chaperone for correct folding of the envelope protein and prevents premature fusion during virus egress, as well as playing a role in receptor binding during viral entry [49,50]. Consensus mutations identified after passage at 30 °C were all found in structural proteins, indicating they may be broadly adaptive and facilitate cell-specific binding or attachment independent of temperature. Further studies are required to elucidate the precise role of these mutations in host- and temperature-dependent fitness.

Although consensus-level mutations can have a large impact on viral fitness, the mutant swarm also contributes to WNV fitness, adaptation, and evolutionary trajectories [40,52,53]. Previous studies demonstrated that consensus changes are not independently responsible for adaption, that increases in the mutant swarm size is associated with increased adaptability, and that cooperative interactions between co-infecting variants can increase viral fitness [16,40]. Shared minority changes were identified in the NS1 protein. NS1 is known to activate the Toll-like receptors (TLRs) and modulate immune responses, and evidence suggests that secreted flavivirus NS1 proteins generated during the vertebrate phase increase acquisition by vectors [54]. Our results are consistent with previous studies investigating intrahost variation of naturally circulating WNV which demonstrate significant diversity in NS1 [7,14]. Additionally, minority changes were identified in the 5′ UTR and previous work demonstrates that the 5′ UTR promotes RNA synthesis and mediates long RNA–RNA interactions [55]. Mutations in the 3′ UTR of WNV have been linked to increased expression of derived subgenomic flavivirus RNA (sfRNA) known to determine mosquito transmission [56,57], yet it is not clear if the 5′ UTR could play a similar role in RNAi regulation. While we did not identify temperature-specific differences in the accumulation of consensus mutations, we did identify increased nonconsensus diversity at higher temperatures, which was significantly higher for nonsynonymous mutations. Although the specific genetic signatures that result in broader adaptation at higher temperatures are not entirely clear, increased amino acid change is consistent with this phenotype and could ultimately result in more frequent fixation of higher fitness mutations over longer temporal scales.

An additional consideration is that the diurnal temperature range can impact arboviral transmission and will fluctuate based on geographical location and season [58]. Since our studies were completed with constant temperatures, it is not clear if fluctuations around different means would result in similar adaptive and evolutionary pressures. Additionally, because WNV cycles between avian and mosquito hosts, future studies should investigate fitness in both vertebrate hosts and mosquitoes at increased temperatures. Such experiments would expose the virus to host-specific selective pressures which may lead to increased purifying selection [13]. While not statistically significant, our data do demonstrate decreased growth rates in avian cells that could be further exacerbated by more extensive adaptation to mosquito cells [59,60]. Genetic drift resulting from bottlenecks and founder effects are also important considerations, in the study of both natural and experimental systems. While our study does not consider the bottlenecks imposed on WNV populations with mosquito infection, dissemination, and transmission [19], bottlenecks from sequential in vitro passage will additionally influence evolutionary trajectories and can often impact fitness [61]. The population sizes used here did not impose the severe bottlenecks historically associated with significant decreases in fitness [62] yet we did observe a decrease in fitness early in passage at the lower temperature, suggesting that the effect of temperature on the capacity of viral populations to overcome bottlenecks is an additional consideration.

Overall, these data demonstrate that failure to consider changing evolutionary trajectories when forecasting the possible effects of climate change on arbovirus transmission likely underestimates the effects of rising temperatures. If evolution at higher temperatures facilitates broad adaption then rising temperatures in nature could promote increases in prevalence, geographic distribution, and vector range. Future work should aim to investigate the role of temperature more thoroughly in natural systems and utilize these data to inform more accurate predictive models.

## Figures and Tables

**Figure 1 viruses-13-01889-f001:**
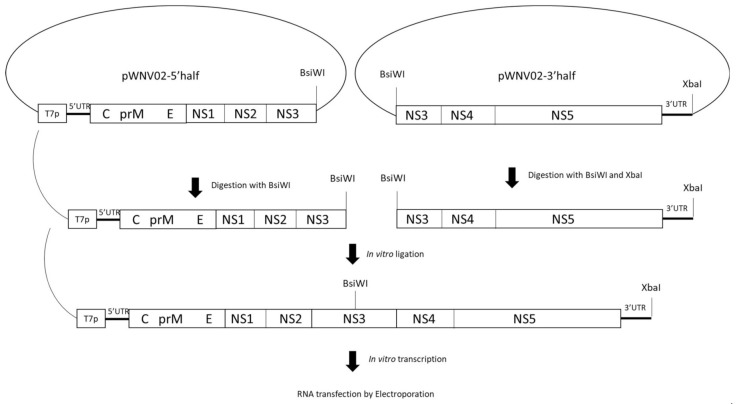
Generation of a WNV02 infectious clone. A low-copy number plasmid (pACYC177) was modified to include BamHI, BsiWI, and XbaI restriction sites. The viral cDNA was cloned into two fragments, pWNV02-5′ half and pWNV02-3′ half. Plasmid pWNV02-5′ half contains the T7 promoter and the 5′-5785 nucleotides of the viral genome that were cloned between BamHI and BsiWI sites, and pWNV02-3′ half contains the 3′-distal 5244 nucleotides of the viral genome between BsiWI and XbaI. Digested and ligated plasmids result in linearized cDNA of the full genome. The full-length genome was purified followed by in vitro transcription and transfection via electroporation.

**Figure 2 viruses-13-01889-f002:**
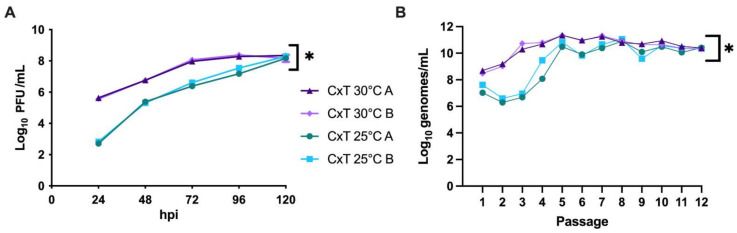
Temperature-dependent replication of West Nile virus in Culex cell culture. (**A**) Growth kinetics of WNV02 IC passage 1 in Culex tarsalis (C × T) cells at 25 °C and 30 °C. Viral loads were determined via plaque titration and kinetics are shown as means of duplicate assay. Significantly higher output titers were measured at 30 °C relative to 25 °C (* *p* < 0.05 (one-way ANOVA w/multiple comparisons, Tukey’s post-test). (**B**) Output viral load during passage of WNV02 IC. Viral RNA levels were quantified at 96 h post-infection by real-time RT-PCR using genome copy standards. Significantly higher output titers were measured at 30 °C relative to 25 °C (* *p*< 0.05 (one-way ANOVA w/multiple comparisons, Tukey’s post-test).

**Figure 3 viruses-13-01889-f003:**
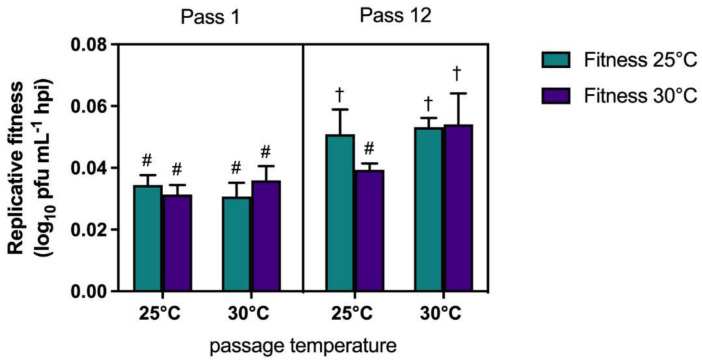
West Nile virus passage at increased temperatures facilitates temperature independent adaptation. Replicative fitness of WNV02 IC was assessed after 1 and 12 passages in Culex tarsalis (C × T) cells. Replicative fitness refers to the slope of the line of best fit determined via linear regression analysis of the log linear portion of individual growth curves. Data are shown as means of duplicate assays and lineages. Statistically different measures are indicated by unique characters (*p* ≤ 0.05, three-way ANOVA w/multiple comparisons, Tukey’s post-test).

**Figure 4 viruses-13-01889-f004:**
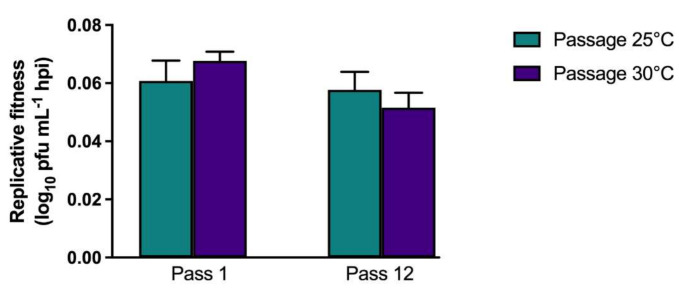
West Nile virus adaptation resulting from passage on mosquito cell culture is host specific. Replicative fitness in avian (PDE) cells of WNV02 IC after passaging in *Culex tarsalis* (C × T) cells. Replicative fitness is shown as means of replicate assay and lineage. No significant differences were measured when comparing passage number or temperatures (*p* > 0.05, one-way ANOVA w/multiple comparisons, Tukey’s post-test).

**Figure 5 viruses-13-01889-f005:**
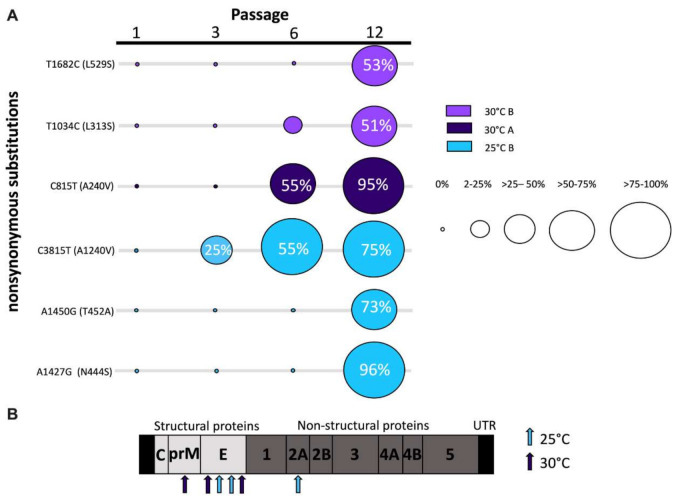
Nonsynonymous consensus mutations identified in West Nile virus following passage in mosquito cells. (**A**) Frequency of fixed nonsynonymous mutations identified throughout passaging. The proportion of sequences with mutation present is indicated by circle radius. Color indicates temperature and lineage. (**B**) Schematic of WNV genome and amino acid substitutions. Arrows indicate amino acid substitution location and color indicates passage temperature of associated mutations.

**Figure 6 viruses-13-01889-f006:**
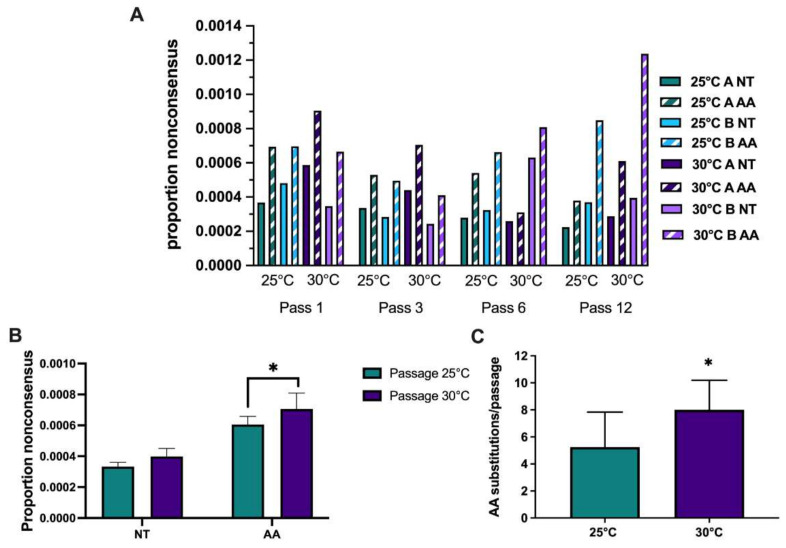
Increased nonsynonymous minority mutations identified in West Nile virus populations passaged at higher temperatures on mosquito cells. Deep sequencing was completed on the Illumina MiSeq platform and analyses were completed using Geneious R11 and MEGA X. (**A**) The proportions of nonconsensus nucleotide and amino acid variants >2% in each lineage during passage at either 25 °C or 30 °C are shown. Solid bars indicate nonconsensus nucleotide (NT) mutations and stripes indicate nonconsensus amino acid (AA) substitutions. (**B**) Mean proportions of nonconsensus nucleotide and amino acid variants identified at frequencies >2% during passage at 25 °C or 30 °C. A significantly higher proportion of nonconsensus amino acid substitutions was identified during WNV passage at 30 °C (* *p* < 0.05, chi-square test). (**C**) A significantly higher proportion of unique amino acid substitutions at >5% per passage was identified in WNV populations passaged at 30 °C (* *p* ≤ 0.05, *t*-test).

**Table 1 viruses-13-01889-t001:** Minority nucleotide (NT) and amino acid (AA) changes in >3% of West Nile virus sequences following 12 passages in *Culex tarsalis* (C × T) cells at 25 °C or 30 °C.

NT Position	Temp and Lineage	Gene	AA Change
134	25 B	5′ UTR	None
139	25 B	5′ UTR	None
144	25 B	5’ UTR	None
151	25 A, 25 B, 30 B	5’ UTR	None
158	25 B	5’ UTR	None
167	25 B	5’ UTR	None
187	25 A	C	K -> E
201	25 B	C	L -> F
242	25 B	C	A -> G
248	25B	C	L -> S
265	25 A	C	T -> A
647	30 B	prM	D -> G
749	30 B	prM	T -> I
751	25 A	prM	V -> M
815	30 B	prM	A -> V
897	30 B	prM	M -> I
898	30 B	prM	L -> A
1114	30 B	E	A -> T
1142	30 A, 30 B	E	Y -> C
1261	25 B	E	R -> G
1343	25 B	E	I -> T
1367	30 A	E	N -> T
1412	30 B	E	V -> A
1484	30 A	E	A -> V
1567	30 B	E	Y -> H
1642	30 B	E	S -> G
1682	30 B	E	X -> S
1801	30 B	E	V -> L
1831	25 A	E	R-> G
1906	30 A	E	T-> A
2190	30 B	E	F -> L
2573	30 B	E	Y -> C
2810	25 B	E	G -> V
2816	25 B	NS1	K -> R
2998	25 B	NS1	T -> P
3234	25 A, 25 B	NS1	N -> K
3287	25 A, 25 B, 30 B	NS1	V -> G
3325	25 A, 25 B, 30 A, 30 B	NS1	T -> P
3333	25 B, 30 B	NS1	S -> R
3337	25 A, 25 B, 30 B	NS1	S -> G
3340	25 B	NS1	C -> G
3389	30 B	NS1	L -> S
3859	25 B	NS1	V -> F
4297	30 B	NS2A	D -> N
4351	30 B	NS2A	I -> L
4682	30 A	NS2A	R -> K
5449	30 A	NS3	N -> D
5755	30 A	NS3	K -> E
6347	25 B, 30 A	NS3	E -> G
6556	30 A	NS3	V -> M
8181	30 A	NS4B	M -> I
8399	25 A	NS4B	L -> P
8833	25 A	NS5	A -> T
9382	25 A	NS5	R -> G
9637	30 A	NS5	F -> V

## Data Availability

WGS data were deposited in the GenBank database, accession numbers are as follows: MZ595324, MZ595325, MZ595326, MZ595327, MZ595328, MZ595329, MZ595330, MZ595331, MZ595332, MZ595333, MZ595334, MZ595335, MZ595336, MZ595337, MZ595338, MZ595339.

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
