# Peer review of "Experimental Evolution of West Nile Virus at Higher Temperatures Facilitates Broad Adaptation and Increased Genetic Diversity"

_viruses, 2021, doi:10.3390/v13101889_

Round 1

Reviewer 1 Report

In their work, Fay and co-workers investigate the effect that the increase in temperature could have on the rate and existence of evolution in flaviviruses, most of which are primarily transmitted to their vertebrate host by the bite from an infected arthropod. To do this, authors infected Culex tarsalis cells with a WNV strain RNA (amplificated in C6 / 36 cells), passaged 12 times at 25° and 30°C in two independent lines. The authors found mutation signatures using full-genome deep sequencing analysis on the viral genomes obtained at the 1th, 3rd, 6th and 12th passage. To assess if passage and adaptation to mosquito cells resulted in altered growth kinetics in avian cells, Fay and co-workers also infected PDE cells with viruses that infected mosquitoes cells at 25° and 30 ° and quantified their fitness replicate. What they claim to get from their results is that the increases in temperature could accelerate emergence of high fitness strains.

In general, I find that this is a good job, interesting and suitable for publication in this Journal. I would have only minor comments (which in part only serve to clarify my doubts due to gaps in some fields of Fay & co-workers research).

Minor Comments:

1-in the abstract, line 18-19: perhaps saying that this experiment investigates "how temperature impacts the rate and extent of flavivirus evolution in mosquito cells" is a bit risky. Flaviviruses are very different from each other, West Nile virus possibly can be considered a representative of a part of flaviviruses (mosquito-borne)

2- I’m not an expert in infection experiments. I would like some steps to be better explained.

For example, what is meant by “the maximum viral imput was utilized” in line 126 of the materials and methods?

25 ° C is a lower T than which mosquito cells generally grow (28 ° C). Given that, even in the discussion, it is said that "unimodal relationships based on experimental data for WNV suggest that although transmission efficiency peaks between 23 ° C and 26 ° C, replication rates in mosquitoes likely peak at temperatures between 35 ° C and 40 ° C", and that "RNAi has been shown to be influenced by temperature, with reduced activity at cooler temperatures. Given that A1240V in the NS2A was associated with temperature-specific adaptation at 25 ° C, it is feasible that this residue could have a role in RNAi suppression. temperature” wouldn't it also be appropriate to use as a control what also happens to the cells passeged to a control T of 28 °? If cells that usually grow at 28 ° C are grown at 25 ° C, can't what happens at 25 ° C be considered more of an effect of lowering the temperature?

3- Figure 2B (and related results). The greatest variability occurs in the first steps: it would not be worth investigating (comment more on the results in the discussion) why this happens?

4- line 213: it is not clear to me why the replication fitness for the 30 ° passage is evaluated only in the A line.

Paragraph 3.2 is a bit difficult to follow, perhaps it should be made smoother. It is not clear to me if the figure shows only steps 1 and 12, but in reality the same analyses were carried out at all steps (or steps 3 and 6), but they were not shown. Also it is not clear to me why to do pass 1 it was chosen to show line A both at 25° and 30 ° C and for pass 12 was used the line B except for fitness at 30 ° C. Line A and line B grown at the same T are expected to have similar trends. If so, isn't it worth showing the results as an average of lines A and B? If, on the other hand, this is not the case, then it would not be worth showing for each case what happens for both lines A and B (for example: PASS1 → A 25 ° vs A30 ° at 25 °; A25 ° vs A30 ° at 30 °; B 25 ° vs B30 ° at 25 °; B25 ° vs B30 ° at 30 ° ,; and the same for PASS12). Or perhaps the same line should be chosen for both steps. Given that the greatest variability occurred in the first steps, wouldn't it be useful to hypothesize what happens also in these steps?

5- line 228: the experiment in PDEs should perhaps be discussed more, otherwise it could also be removed.

6- Figure 5. It is still not clear to me why only one line is considered for the 25 °C passage.

Excluding these minor revisions, I find this to be a good and very interesting work, suitable for publication in this journal. I find it would also be interesting to evaluate the same effect of temperature on vertebrate cells.

Author Response

In their work, Fay and co-workers investigate the effect that the increase in temperature could have on the rate and existence of evolution in flaviviruses, most of which are primarily transmitted to their vertebrate host by the bite from an infected arthropod. To do this, authors infected Culex tarsalis cells with a WNV strain RNA (amplificated in C6 / 36 cells), passaged 12 times at 25° and 30°C in two independent lines. The authors found mutation signatures using full-genome deep sequencing analysis on the viral genomes obtained at the 1th, 3rd, 6th and 12th passage. To assess if passage and adaptation to mosquito cells resulted in altered growth kinetics in avian cells, Fay and co-workers also infected PDE cells with viruses that infected mosquitoes cells at 25° and 30 ° and quantified their fitness replicate. What they claim to get from their results is that the increases in temperature could accelerate emergence of high fitness strains. In general, I find that this is a good job, interesting and suitable for publication in this Journal. I would have only minor comments (which in part only serve to clarify my doubts due to gaps in some fields of Fay & co-workers research)

In the abstract, line 18-19: perhaps saying that this experiment investigates "how temperature impacts the rate and extent of flavivirus evolution in mosquito cells" is a bit risky. Flaviviruses are very different from each other, West Nile virus possibly can be considered a representative of a part of flaviviruses (mosquito-borne)

Response: We agree with the reviewer’s opinion that WNV is not necessarily representative of all flaviviruses and have edited the abstract appropriately (line 19).

I’m not an expert in infection experiments. I would like some steps to be better explained.

 For example, what is meant by “the maximum viral imput was utilized” in line 126 of the materials and methods?

Response: Thank you for pointing out that oversight, we recognize that we neglected to point out when the maximum viral input was utilized. To clarify, if the titer was not high enough to be diluted then the virus was blind passed. This only occurred once during passage temperature 25°C lineage A, pass 3, where cells were inoculated using an MOI of 0.008 . We added a section to the methods to more thoroughly explain (Line 126). Additionally, we have added a section to the discussion about the influence of bottlenecks on viral fitness and evolution (lines 393-402).

25 ° C is a lower T than which mosquito cells generally grow (28 ° C). Given that, even in the discussion, it is said that "unimodal relationships based on experimental data for WNV suggest that although transmission efficiency peaks between 23 ° C and 26 ° C, replication rates in mosquitoes likely peak at temperatures between 35 ° C and 40 ° C", and that "RNAi has been shown to be influenced by temperature, with reduced activity at cooler temperatures. Given that A1240V in the NS2A was associated with temperature-specific adaptation at 25 ° C, it is feasible that this residue could have a role in RNAi suppression. temperature” wouldn't it also be appropriate to use as a control what also happens to the cells passaged to a control T of 28 °? If cells that usually grow at 28 ° C are grown at 25 ° C, can't what happens at 25 ° C be considered more of an effect of lowering the temperature?

Response: While these cells are normally maintained at 28C, they are viable and healthy within the range of 25-30C. It also should be clarified that cells were never passaged, only supernatant and the virus has not previously been adapted to 28C. That said, we agree with the reviewer’s comment that growth at 25°C and 28°C may induce different RNAi responses resulting from the effect of cell temperature rather than viral mutations. This is indeed the point we are trying to make, that temperature specific viral mutations could feasibly be a response to differential, temperature-dependent cellular responses. We assume using larger differences (25 v 30C) would highlight this effect, yet it is possible that responses could be different at 28C. Further studies at a range of temperatures would be required to effectively measure this but we feel this study is provides a good baseline to expand to investigation of the mechanistic basis of adaptation at a range of temperatures.

Figure 2B (and related results). The greatest variability occurs in the first steps: it would not be worth investigating (comment more on the results in the discussion) why this happens?

perhaps the same line should be chosen for both steps. Given that the greatest variability occurred in the first steps, wouldn't it be useful to hypothesize what happens also in these steps?

Response: There are a number of examples where repeated passage of RNA viruses in culture initially results in a decrease in fitness. This could be attributed to accumulation of non-infectious particles or repeated bottlenecks. Although the bottlenecks were not so tight that we would anticipate a significant ratcheting effect, this is a possibility that we have now addressed in the discussion (lines 393-402). We were also intrigued by the variability in titers during early passages which led us to genetically characterize passage 3. The results showed that there was little change on the consensus level at pass 3. The broader goal of this study was to investigate the potential longer-term effects of temperature on viral evolution, which is why we focused on more comprehensive characterization following the complete passage series.

line 213: it is not clear to me why the replication fitness for the 30 ° passage is evaluated only in the A line.  

Paragraph 3.2 is a bit difficult to follow, perhaps it should be made smoother. It is not clear to me if the figure shows only steps 1 and 12, but in reality the same analyses were carried out at all steps (or steps 3 and 6), but they were not shown. Also it is not clear to me why to do pass 1 it was chosen to show line A both at 25° and 30 ° C and for pass 12 was used the line B except for fitness at 30 ° C. Line A and line B grown at the same T are expected to have similar trends. If so, isn't it worth showing the results as an average of lines A and B? If, on the other hand, this is not the case, then it would not be worth showing for each case what happens for both lines A and B (for example: PASS1 → A 25 ° vs A30 ° at 25 °; A25 ° vs A30 ° at 30 °; B 25 ° vs B30 ° at 25 °; B25 ° vs B30 ° at 30 ° ,; and the same for PASS12).

Response: Both passage 1 and 12 for both lineages were indeed evaluated at both temperatures.  We regret the confusion and acknowledge that figure 3 can be difficult to follow. We have tried to clarify with some minor changes. While we outline the fitness of each lineage at each temperature in the results section, the figure is showing the average replicative fitness of lineage A and B, as the reviewer recommends. We have clarified the results (line 210) and, rather than using A and B to signify significance, we are now using symbols in Fig 3. Given that lineages are also designated A and B we believe these designations were confusing to the reader.

 line 228: the experiment in PDEs should perhaps be discussed more, otherwise it could also be removed.                                                                                                                                                               Response: We agree with the reviewer that the discussion was lacking follow up on the PDE results shown in figure 4. We have elaborated on the implications of the PDE results (line 387-393).

Figure 5. It is still not clear to me why only one line is considered for the 25 °C passage.                        Response: Thank you for this comment. To clarify, this figure is only showing lineages that consensus changes were identified within. There were no consensus changes found in lineage A at passage temperature 25°C thus it is not shown.

Excluding these minor revisions, I find this to be a good and very interesting work, suitable for publication in this journal. I find it would also be interesting to evaluate the same effect of temperature on vertebrate cells.                                                                                                                          Response: Thank you for your positive feedback. We agree that similar experiments in vertebrate cells would be interesting but, given that mosquitoes are ectothermic, the virus is subjected to changes in external temperature only in the vector, so we chose to focus on this system initially.

Reviewer 2 Report

The paper is well designed and the results are nicely presented. In general is a good effort. I have some minor suggestions regarding the data presentation: 

1. Please avoid using red in your figures, so colour blind readers can clearly read your results. 2. The asterisks in figure 2 are misplaced, resulting in confusion. 3. In figure 5, I suggest percentages are shown in the middle for largest cycles.

Author Response

The paper is well designed and the results are nicely presented. In general is a good effort. I have some minor suggestions regarding the data presentation:

Please avoid using red in your figures, so colour blind readers can clearly read your results. 2. The asterisks in figure 2 are misplaced, resulting in confusion. 3. In figure 5, I suggest percentages are shown in the middle for largest cycles.

Response: Thank you for the suggestions. We have changed the colors throughout and added percentages to figure 5. Regarding the asterisks used in figure 2, they are showing that the lines are each statistically significantly different from each other, not individual points, so the placement is accurate.

Reviewer 3 Report

Fay et al. investigate experimental evolution of West Nile virus under different temperature regimes in Culex tarsalis cells. WNV was passaged 12 times at either 25C or 30C, with two biological replicates each. Replicative fitness was determined in mosquito and avian cells at both temperatures, and genetic diversity was determined by NGS. The authors study an interesting fundamental question regarding the effect of temperature on WNV evolution. The results are clearly presented and the manuscript is well written, but the overall implications of the study are unclear and conclusions are not fully supported by the data.

  1. Passaging of virus under the two different temperature regimes were performed with 2 biological replicates. Although overall growth kinetics were consistent between the two replicates, there were inconsistencies between replicates when replicative fitness was determined at matched and alternative temperatures. Moreover, inconsistencies between replicates were also found when comparing genetic diversity. The authors conclude that increasing temperatures could facilitate the emergence of broadly adaptive, high fitness WNV strains. However, given the inconsistencies despite the 1) highly controlled experimental conditions, 2) constant temperatures (instead of fluctuating temperatures in nature), 3) mosquito cell model system that lacks bottlenecks that are present in live mosquitoes, and 4) no alternation between mosquito and avian host, I would argue that an alternative conclusion could be that there is no clear evidence for adaptation to higher temperatures. Instead of adaptation to temperature, bottlenecks in passaging and drift might be more important factors that may explain the differences between replicates. The authors should be careful in their overall conclusions and discussion of implications for the field.

  1. Minority genetic variation was investigated at a threshold of 2% at 100X coverage, without technical replicates, and results were averaged among 12 passages for each temperature. Previous studies have provided guidelines for measuring intrahost variation which includes normalized RNA input into cDNA synthesis (at least 1000 RNA copies), preparing libraries in duplicate, sequence to a minimum depth of coverage of 400X, and only calling true iSNV at greater than 3% frequency that are present in both replicates (https://doi.org/10.1186/s13059-018-1618-7). The current approach deviates from these recommendations and therefore the authors should be very careful when interpreting these results. Moreover, results should not be averaged across passages, but rather presented for each passage to show changes over time. If possible, it would be desirable to repeat the sequencing to reduce the possible calling of false SNPs. If this is not possible, the authors should provide (reference to) validation of their approach.

  1. The MOI was kept constant between passages, but the authors report that the maximum viral input was utilized when the desired MOI was not achieved. How often was the MOI not achieved, and how different was this between lineages and temperatures? This information should be included in the supporting data. Changes in MOI as well as changes in peak titers change the size of the bottleneck between passages. Potential implications of differences in bottleneck between the two temperatures (e.g. initially higher titers at 30C and, thus, larger bottleneck between passages) should be discussed.

Author Response

Fay et al. investigate experimental evolution of West Nile virus under different temperature regimes in Culex tarsalis cells. WNV was passaged 12 times at either 25C or 30C, with two biological replicates each. Replicative fitness was determined in mosquito and avian cells at both temperatures, and genetic diversity was determined by NGS. The authors study an interesting fundamental question regarding the effect of temperature on WNV evolution. The results are clearly presented and the manuscript is well written, but the overall implications of the study are unclear and conclusions are not fully supported by the data.

Passaging of virus under the two different temperature regimes were performed with 2 biological replicates. Although overall growth kinetics were consistent between the two replicates, there were inconsistencies between replicates when replicative fitness was determined at matched and alternative temperatures. Moreover, inconsistencies between replicates were also found when comparing genetic diversity. The authors conclude that increasing temperatures could facilitate the emergence of broadly adaptive, high fitness WNV strains. However, given the inconsistencies despite the 1) highly controlled experimental conditions, 2) constant temperatures (instead of fluctuating temperatures in nature), 3) mosquito cell model system that lacks bottlenecks that are present in live mosquitoes, and 4) no alternation between mosquito and avian host, I would argue that an alternative conclusion could be that there is no clear evidence for adaptation to higher temperatures. Instead of adaptation to temperature, bottlenecks in passaging and drift might be more important factors that may explain the differences between replicates. The authors should be careful in their overall conclusions and discussion of implications for the field.

Response: We appreciate the comment and certainly agree that other forces could be at play here. We do stand by the interpretation that our results support the idea that increased temperature increases the probability of the emergence of broadly adaptive strains for the following reasons: (1) While the magnitude was different, we measured increased replicative fitness for both lineages passed at the higher temperature at both 25 and 30C.  More inconsistency was measured at the lower temperature, which is consistent with our interpretation and (2) On average we see more mutation, particularly on the amino acid level, at higher temperatures. We have qualified our interpretation to acknowledge the potential role stochastic forces in the discussion (lines 393-402) and certainly agree that there is much more complexity to consider for strains evolving in natural transmission cycles (see lines 384-409). 

 Minority genetic variation was investigated at a threshold of 2% at 100X coverage, without technical replicates, and results were averaged among 12 passages for each temperature. Previous studies have provided guidelines for measuring intrahost variation which includes normalized RNA input into cDNA synthesis (at least 1000 RNA copies), preparing libraries in duplicate, sequence to a minimum depth of coverage of 400X, and only calling true iSNV at greater than 3% frequency that are present in both replicates (https://doi.org/10.1186/s13059-018-1618-7). The current approach deviates from these recommendations and therefore the authors should be very careful when interpreting these results. Moreover, results should not be averaged across passages, but rather presented for each passage to show changes over time. If possible, it would be desirable to repeat the sequencing to reduce the possible calling of false SNPs. If this is not possible, the authors should provide (reference to) validation of their approach.

 Response: We appreciate the reviewer directing us to this publication and certainly agree that the probability of process errors must be fully considered in interpretation of minority variant results. In figure 6 we show in panel A the nucleotide and amino acid diversity observed in >2% at all sequenced passages. As suggested by the reviewer, this includes every sequenced passage and both lineages, presented separately. There is an important distinction between reporting individual SNVs and reporting differences in overall diversity. While a variant at 2% may not itself be reliable, overall differences in diversity should indeed by reliable, as samples are treated in the same manner and the process error rate will be similar among samples.  We did additionally present individual changes in table 1 with a 2% cut-off and, based on these comments, we have now amended this cut-off to 3%. Importantly, in figure 6C we show the amino acid substitutions in >5% of all sequenced passages, a much more conservative cut-off than that is suggested by the reviewer.  The variability among passages, lineages and temperatures is evident in the data presented, but the point in showing the average is to indicate that the probability of accumulating more diversity at higher temperatures is increased, which is consistent with the phenotypic results and among the primary points of the study.

The MOI was kept constant between passages, but the authors report that the maximum viral input was utilized when the desired MOI was not achieved. How often was the MOI not achieved, and how different was this between lineages and temperatures? This information should be included in the supporting data. Changes in MOI as well as changes in peak titers change the size of the bottleneck between passages. Potential implications of differences in bottleneck between the two temperatures (e.g. initially higher titers at 30C and, thus, larger bottleneck between passages) should be discussed.

Response: We certainly fully agree with the critique and regret that this was not clearer. Fortunately, the use of a modestly lower MOI only occurred once during passage temperature 25°C lineage A, pass 3, for which cells were inoculated using an MOI of 0.008 . We added a section to the methods to more thoroughly explain the methods used (Line 127). Additionally, a section was added to the discussion about the influence of bottlenecks on viral fitness (lines 393-402).

Round 2

Reviewer 3 Report

I would like to thank the authors for their responses to my comments and for revising the paper. Although the authors have addressed caveats of the study and its implications, I feel that the overall conclusion is still overstated. The authors investigated replicative fitness and genetic diversity of viruses passaged in mosquito cells, which cannot be directly translated to potential emergence of high fitness strains in nature. Although there is more diversity at higher temperature, there is still a lack of consistent changes (in specific mutations as well as level of genetic diversity between replicates), and implications were not evaluated in live mosquitoes (only in cells). The conclusions should, therefore, be phrased more specific for what was investigated in this study. See below for some specific suggestions:

Title: Experimental evolution of West Nile virus at higher temperatures results in increased replicative fitness and genetic diversity in mosquito cells

Abstract: Remove final sentence (lines 25-26)

Discussion: remove or rephrase final sentence of first paragraph (lines 313-314: suggesting that increasing temperatures in nature…), and lines 425-426 in final paragraph.

Lastly, the authors indicated that they increased their threshold for individual changes to 3%, but I do not see this reflected in other sections of the manuscript. The methods (lines 162-164, 279-280) and figure 6 legend still indicate 2% as a cut-off value.
